# Coupling of Pyro–Piezo-Phototronic Effects in a GaN Nanowire

**DOI:** 10.3390/ma16186247

**Published:** 2023-09-17

**Authors:** Guoshuai Qin, Zhenyu Wang, Lei Wang, Kun Yang, Minghao Zhao, Chunsheng Lu

**Affiliations:** 1School of Electromechanical Engineering, Henan University of Technology, Zhengzhou 450001, China; wangzhenyu962022@163.com; 2Henan Institute of Metrology, Zhengzhou 450001, China; wanglei0404_wl@163.com; 3School of Mechanics and Safety Engineering, Zhengzhou University, Zhengzhou 450001, China; 18838967167@163.com (K.Y.);; 4School of Mechanical Engineering, Zhengzhou University, Zhengzhou 450001, China; 5Henan Key Engineering Laboratory for Anti-Fatigue Manufacturing Technology, Zhengzhou University, Zhengzhou 450001, China; 6School of Civil and Mechanical Engineering, Curtin University, Perth, WA 6845, Australia

**Keywords:** pyro–piezo-phototronic effects, coupling, multi-field, GaN nanowires

## Abstract

In this paper, we systematically investigate the synergistic regulation of ultraviolet and mechanical loading on the electromechanical behavior of a GaN nanowire. The distributions of polarization charge, potential, carriers, and electric field in the GaN nanowire are analytically represented by using a one-dimensional model that combines pyro-phototronic and piezo-phototronic properties, and then, the electrical transmission characteristics are analyzed. The results suggest that, due to the pyro-phototronic effect and ultraviolet photoexcited non-equilibrium carriers, the electrical behavior of a nano-Schottky junction can be modulate by ultraviolet light. This provides a new method for the function improvement and performance regulation of intelligent optoelectronic nano-Schottky devices.

## 1. Introduction

Piezoelectric semiconductors (PSCs) have attracted research attention in the fields of ultraviolet detection, optical recognition device switch and sensor technology [1]. Due to their multi-field coupling characteristics, PSCs can convert multiple physical field signals, such as force [2,3,4,5], electricity [6,7], and light [8,9,10], into electrical signals. In 2010, Wang [11] provided the concept of piezo-phototronics, which deploys the piezopotential caused by strain to regulate the recombination of carriers and improve the efficiency of optoelectronic devices [12,13,14]. It can be utilized to improve the performance and sensitivity of devices like photodetectors and solar cells [15,16,17,18]. Multidimensional optoelectronic modulation has been accomplished by employing the coupling of multiple interfacial physical effects, e.g., pyro-phototronic, piezo-phototronic, and ferroelectricity [19,20,21].

Nonetheless, the existing research predominantly concentrates on the influence of mechanical loading [22] or doping [23] on the electrical transport properties of PSCs. However, based on the best of our knowledge, there is still a huge lack of research addressing the co-regulation of electromechanical behaviors resulting from light stimulation, despite the well-known influence of irradiation on carrier generation, separation, transport, and recombination processes in PSCs [24]. Because of their pyro-phototronic and piezo-phototronic coupling characteristics and high mobility properties [25], GaN nanostructures have obvious advantages in micro-electromechanical systems and active control applications. Thus, GaN nanowire is taken as a research object to reveal the complex coupling mechanism among mechanical, electrical, and optical fields. Owning to the special coupling ability, the piezoelectric potential generated in a crystal can effectively regulate the carrier transport capacity in an interface/unction region under mechanical loading. Such a unique synergistic effect makes the piezoelectric potential produce a similar “gate circuit” [26], which holds the potential to yield significant advancements in the functional development and performance regulation of PSC devices.

This work mainly focuses on the analytical solution of a one-dimensional (1D) GaN nanowire under thermo-optoelectronic and piezoelectric photoelectron coupling. Moreover, the comprehensive regulation of current characteristics is studied using a numerical method. The structure of this paper is outlined as follows: In Section 2, the 1D PSC theory is presented, which encompasses the consideration of photoconductive and photothermal effects. Subsequently, in Section 3, the study delves into the examination of the impact of the synergistic effects of ultraviolet light and compressive stress on the physical fields within the GaN nanowires. Moving on to Section 4, the focus is on discussing the combined effect of ultraviolet and stress regulation on the electrical transport properties of Ag-GaN Schottky structure. Finally, the paper concludes with a summary of the main findings and conclusions in Section 5.

## 2. Basic Equations for a GaN Nanowire

In semiconductors, electron–hole pairs can be generated and excited from the valence band to the conduction band when the energy carried by photons exceeds the band gap energy. The changes of carrier concentration under steady state conditions are mathematically described by
(1)$$\begin{array}{l} \Delta n_{\mathrm{opt}}=\beta \alpha P_{\mathrm{opt}}\lambda h^{-1}c^{-1}e^{-\alpha d}\tau_{n},\\ \Delta p_{\mathrm{opt}}=\beta \alpha P_{\mathrm{opt}}\lambda h^{-1}c^{-1}e^{-\alpha d}\tau_{p}, \end{array}$$
where *β* is quantum efficiency; *α* is the absorption coefficient; *P*_opt_ and *λ* are the illumination intensity and wavelength, respectively. *h*, *c,* and *d* are the Planck constant, light velocity, and incident depth, respectively. *τ_n_* and *τ_p_* are the lifetimes of the excited electron and hole, respectively [24].

As illustrated in Figure 1, for a nanowire with a radius of *r*, the population of photo-excited carriers diminishes with an increasing transmission depth of the incident light. The average photogenerated carrier concentration on the cross-section can be mathematically represented as
(2)$$\begin{array}{l} \Delta {\bar{n}}_{\mathrm{opt}}=(\iint_{x^{2}+y^{2}\leq r^{2}}\Delta n_{\mathrm{opt}}dxdy)/\pi r^{2},\\ \Delta {\bar{p}}_{\mathrm{opt}}=(\iint_{x^{2}+y^{2}\leq r^{2}}\Delta p_{\mathrm{opt}}dxdy)/\pi r^{2}. \end{array}$$

During the relaxation time (~10^−12^ s), photons with energies exceeding the bandgap width can induce lattice vibrations, subsequently converting the energy into thermal energy [27]. As a consequence, a temperature increase occurs in the semiconductor structure. In accordance with the principle of heat balance, the variation of temperature with respect to time can be described as follows [28]:(3)$$C_{T}\frac{d(\theta -\theta_{p})}{dt}+G_{T}(\theta -\theta_{0})=SI,$$
where *C_T_* is the heat capacity and *G_T_* = *H*_g_ *A*_g_ denotes the heat exchange coefficient, with *H*_g_ = 4 WK^−1^ m^−2^ being the air thermal convection coefficient, and *A*_g_ the heat intersection area of surrounding air [20]. *θ_p_* and *θ*_0_ correspond to the initial and room temperatures, respectively. *S* represents the illumination area, and the excess light intensity is denoted by *I* = (*hv* − *qEg*)/(*hv*)*P*_opt_, with *hv* representing the photon energy, *q* the elementary charge, and *Eg* the bandgap energy.

Initially, the temperature change is assumed to be zero, i.e., Δ*θ*(0) = 0. By solving Equation (3), the resulting expression for the temperature change over time is obtained as follows:(4)$$\Delta \theta (t)=\theta -\theta_{p}=SI/G_{\theta }(1-e^{-t/\tau_{\theta }}),$$
where *τ_θ_* and *G_θ_* are the maximum change of temperature and thermal time constant. Here, when the illumination time *t* ≫ *τ_θ_*, we have Δ*θ*(*t*) = Δ*θ*_opt_ = *SI/G_θ_* [28].

The physical and mechanical behaviors of a 1D GaN nanorod with a length of 2*L* (as illustrated in Figure 2) can be governed by the following equations: the motion equation, electrostatics Gauss’s law, and the current continuity equation [29,30,31]. These equations can be represented as
(5a)$$\begin{array}{l} \frac{\partial \sigma_{zz}}{\partial z}=0,\\ \frac{\partial D_{z}}{\partial z}=q(p-n+N_{D}^{}-N_{A}^{}), \end{array}$$
(5b)$$\begin{array}{l} \frac{\partial J_{z}^{n}}{\partial z}=-qU_{n},\\ \frac{\partial J_{z}^{p}}{\partial z}=qU_{p}, \end{array}$$
where *σ_zz_*, *D_z_*, $J_{z}^{n}$ and $J_{z}^{p}$ are the stress tensor, electric displacement, electron concentration density, and hole current density, respectively. *N*, *p*, *N_D_*, and *N_A_* denote the electron concentrations, hole concentrations, the ionization degrees of donor, and acceptor impurities under light, respectively. For convenience, we denote *n*_0_ = *N_D_*, *p*_0_ = *N_A_*, *n* = *n*_0_ + Δ*n*, and *p* = *p*_0_ + Δ*p*. *q* = 1.602 × 10^−19^ C is the unit charge. Under stable irradiation, free electrons and holes are in dynamic equilibrium., i.e., *U_n_* (the net recombination rates of electrons) = *U_p_
*(the net recombination rates of free holes) = 0.

For a 1D PSC, the constitutive equation in Cartesian coordinates [31,32] can be written as
(6)$$\begin{array}{l} \sigma_{zz}=c_{33}\varepsilon_{zz}-e_{33}E_{z}-{\bar{\lambda }}_{33}\theta ,\\ D_{z}=e_{33}\varepsilon_{zz}+\kappa_{33}E_{z}+{\bar{p}}_{33}\theta ,\\ J_{z}^{n}=qn\mu_{33}^{n}E_{z}+qd_{33}^{n}\frac{\partial n}{\partial z}≅qn_{0}\mu_{33}^{n}E_{z}+qd_{33}^{n}\frac{\partial \Delta n}{\partial z},\\ J_{z}^{p}=qp\mu_{33}^{p}E_{z}-qd_{33}^{p}\frac{\partial p}{\partial z}≅qp_{0}\mu_{33}^{n}E_{z}+qd_{33}^{n}\frac{\partial \Delta p}{\partial z}, \end{array}$$
where *ε_zz_*, *E_z_*, and *κ*_33_ are the strain tensor, electric field strength, and dielectric constant, respectively. ${\bar{\lambda }}_{33}$ and ${\bar{p}}_{33}$ denote the thermal expansion coefficient and pyroelectric coefficient, respectively. *c*_33_ is the elastic coefficient, *e*_33_ is the piezoelectric coefficient, $\mu_{33}^{n}$ and $\mu_{33}^{p}$ denote the electron and hole mobilities, respectively, and $d_{33}^{n}$ and $d_{33}^{p}$ are electron and hole diffusion constants, respectively. The diffusion and mobility of carriers satisfy Einstein relation [33], namely
(7)$$\frac{\mu_{33}^{n}}{d_{33}^{n}}=\frac{\mu_{33}^{p}}{d_{33}^{p}}=\frac{q}{k_{B}T_{0}}.$$
where *k_B_* and *T*_0_ are Boltzmann’s constant and reference temperature. The strain *ε_zz_* is related to the mechanical displacement *u*, and the electric field *E_z_* is related to the electric potential *φ*; that is
(8)$$\begin{array}{l} \varepsilon_{zz}=\frac{\partial u_{z}}{\partial z},\\ E_{z}=-\frac{\partial \varphi }{\partial z}, \end{array}$$
where *u_z_* is the displacement and *φ* is potential.

## 3. Light Irradiation-Induced Electromechanical Fields

The boundary conditions at the two ends can be described as follows under the illustrated ultraviolet light and mechanical conditions in Figure 2, when an applied current flows across the nanorod (in and out):(9a)$$\begin{array}{l} \sigma_{z}(\pm L)=f,\\ D_{z}(\pm L)=0, \end{array}$$
(9b)$$\begin{array}{l} J_{z}^{n}(\pm L)=0,\\ J_{z}^{p}(\pm L)=0. \end{array}$$

Here, Δ*n* and Δ*p* satisfy the electrical neutral conditions, that is
(10)$$\begin{array}{l} \int_{-L}^{L}\Delta pdz=0,\\ \int_{-L}^{L}\Delta ndz=0. \end{array}$$

At the position *z* = 0, the displacement and potential are $u\left(0\right)=0$ and $\varphi \left(0\right)=0$, respectively.

From Equations (5b) and (9b), we can obtain $J_{z}^{n}=0,J_{z}^{p}=0$

Substituting Equation (6) into Equation (5a) and $J_{z}^{n}=0,J_{z}^{p}=0$, the governing equations are obtained as
(11)$$\begin{array}{l} c_{33}\frac{\partial^{2}u}{\partial z^{2}}+e_{33}\frac{\partial^{2}\varphi }{\partial z^{2}}-{\bar{\lambda }}_{33}\frac{\partial \theta }{\partial z}=0,\\ e_{33}\frac{\partial^{2}u}{\partial z^{2}}-\kappa_{33}\frac{\partial^{2}\varphi }{\partial z^{2}}+{\bar{p}}_{33}\frac{\partial \theta }{\partial z}=q(\Delta p-\Delta n),\\ -qn_{0}\mu_{33}^{n}\frac{\partial \varphi }{\Delta z}+qd_{33}^{n}\frac{\partial \Delta n}{\Delta z}=0,\\ -qp_{0}\mu_{33}^{p}\frac{\partial \varphi }{\Delta z}-qd_{33}^{p}\frac{\partial \Delta p}{\Delta z}=0. \end{array}$$

Then, a differential equation about the potential *φ* can be written as
(12)$$\frac{\partial^{3}\varphi }{\partial z^{3}}-\kappa^{2}\frac{\partial \varphi }{\partial z}=0,$$
where $\;k^{2}=q\left(n_{0}+q_{0}\right)V_{th}\varepsilon_{33}^{-1},V_{th}=qK_{B}^{-1}T_{o}\;\mathrm{and}\;\varepsilon_{33}=k_{33}+e_{33}^{2}C_{33}^{-1}$. The general solution is
(13)$$\varphi =C_{1}\cosh\kappa z+C_{2}\sinh\kappa z+C_{3}.$$

Further, based on Equation (13), the general solutions of *u*, *n,* and *p* are
(14)$$u=-\frac{e_{33}}{C_{33}}\varphi +C_{4}z+C_{5}\;\Delta n=n_{0}V_{th}\varphi +C_{6}\;\Delta p=\Delta n-\frac{\varepsilon_{33}}{q}\frac{\partial^{2}\varphi }{\partial z^{2}}$$
where *C_x_* (*x* = 1, 2, …, 6) are the integral constants.

Substituting Equations (13) and (14) into Equations (9a), (10), and $u\left(0\right)=0,\varphi \left(0\right)=0$, the unknown integral constants are determined as
(15)$$\begin{array}{l} C_{1}=0,C_{2}=0,\\ C_{3}=\frac{{\bar{\lambda }}_{33}\theta e_{33}+e_{33}f+c_{33}{\bar{p}}_{33}\theta }{\kappa \varepsilon_{33}c_{33}\cosh\kappa L},\\ C_{4}=\frac{{\bar{\lambda }}_{33}\theta +f}{c_{33}},\\ C_{5}=0,C_{6}=0. \end{array}$$

Thus, the analytical expressions of each electromechanical field are


(16)
$$\begin{array}{l} \varphi =\frac{\left({\bar{\lambda }}_{33}\theta e_{33}+e_{33}f+c_{33}{\bar{p}}_{33}\theta \right)}{\kappa \varepsilon_{33}c_{33}\cosh\kappa L}\sinh\kappa z,\\ u=-\frac{\left({\bar{\lambda }}_{33}\theta e_{33}^{2}+e_{33}^{2}f+e_{33}c_{33}{\bar{p}}_{33}\theta \right)}{\kappa \varepsilon_{33}c_{33}^{2}\cosh\kappa L}\sinh\kappa z+\frac{{\bar{\lambda }}_{33}\theta +f}{c_{33}}z,\\ n=n_{0}+n_{0}V_{th}\frac{\left({\bar{\lambda }}_{33}\theta e_{33}+e_{33}f+c_{33}{\bar{p}}_{33}\theta \right)}{\kappa \varepsilon_{33}c_{33}\cosh\kappa L}\sinh\kappa z,\\ p=p_{0}-p_{0}V_{th}\frac{\left({\bar{\lambda }}_{33}\theta e_{33}+e_{33}f+c_{33}{\bar{p}}_{33}\theta \right)}{\kappa \varepsilon_{33}c_{33}\cosh\kappa L}\sinh\kappa z,\\ \rho =-q(n_{0}+p_{0})V_{th}\frac{\left({\bar{\lambda }}_{33}\theta e_{33}+e_{33}f+c_{33}{\bar{p}}_{33}\theta \right)}{\kappa \varepsilon_{33}c_{33}\cosh\kappa L}\sinh\kappa z,\\ E=-\frac{\left({\bar{\lambda }}_{33}\theta e_{33}+e_{33}f+c_{33}{\bar{p}}_{33}\theta \right)}{\varepsilon_{33}c_{33}\cosh\kappa L}\sinh\kappa z. \end{array}$$


It is worth noting that, such a mathematical model is universal for thermo-optoelectronic and piezoelectric photoelectron coupling, and thus, it can be generalized to other PSCs. In essence, the coupling of mechanical, electric, thermal, and optical fields is realized by the polarized charge generated by each physical field itself, which affects the distribution or flow of carriers. As a preliminary study, the effectiveness of this model has been verified by numerical methods. However, further work is still needed to elucidate the effect of such a complex and coupling field.

Here, based on these analytical expressions, we investigated the synergistic effect of ultraviolet light with a wavelength of 210 nm and a compressive stress on the electromechanical behaviors of PSCs. Let us take an n-type GaN as an example, with its relevant material constants taken from references [34,35,36].

The photoconductive effect leads to a significant generation of photogenerated carriers through light excitation, facilitated by the combined influence of high-energy ultraviolet light and pressure stress, as depicted in Figure 3. These carriers effectively enhance the carrier concentration within the GaN nanorod. Furthermore, the distributions of carriers, polarization charges, potential, and electric field strength are profoundly influenced by the intensity of ultraviolet radiation, particularly at both ends. This is mainly due to both the photoinduced pyroelectric and piezoelectric effects. Since the photon energy of ultraviolet light with a wavelength of 210 nm is higher than the band gap width of GaN, the excess energy can cause lattice vibration and then convert into heat energy. The formal effect induces the segregation of ions within a GaN nanowire, leading to the generation of pyroelectric charges. The polarity of pyro-polarization generated by illumination is opposite to that of piezo-polarization generated by compressive stress. When the UV intensity is low, piezo-electric polarization is dominant. In contrast, with the increase in light intensity, the photo-induced pyroelectric effect gradually affects the distribution of comprehensive polarized charge density (see Figure 4a). Similarly, the redistribution of an effective composite polarization charge causes the corresponding changes of potential and electric fields that decrease with the ultraviolet light intensity, and then reverse in direction and gradually increase (see Figure 4b–d). It is seen that the model can effectively reflect the coupling of pyro–piezo-phototronic effects, which provides a theoretical basis for further understanding the essence of the multi-field coupling of PSCs.

## 4. *I–V* Characteristics under Light and Pressure

Let us take the Schottky (Ag-GaN)-Ohmic contact as an example; the boundary conditions are given by [37,38,39,40]
(17)$$\begin{array}{l} \sigma_{z}(\pm L)=f,\\ V(-L)=V_{a}+V_{b},\\ V(L)=0,\\ u(0)=0,\\ J_{z}^{n}(-L)=-qv_{\mathrm{rec}}^{n}(n-n_{m}),\\ J_{z}^{p}(-L)=qv_{\mathrm{rec}}^{p}(p-p_{m}),\\ n(L)=N_{D},\\ p(L)=N_{A}. \end{array}$$
where *V*_a_ is an applied bias voltage and the effective built-in voltage *V*_b_ = *V*_bi_ + *V*_ph_ + *V*_pizeo_ + *V*_pyro_. *V*_bi_ is the built-in voltage without load. *V*_ph_ [14], *V*_pizeo_, and *V*_pyro_ are the potential differences between the two ends caused by photogenerated carriers, piezo-charges, and pyro-charges, respectively.

In consideration of the nonlinear behavior observed in typical I–V curves of Schottky junctions, a numerical iterative method is employed using the nonlinear version of constitutive relations and the PDE module in COMSOL Multi-physics (Version 5.5) software. As depicted in Figure 4, the synergistic effect of pyro-phototronic and piezo-phototronic phenomena can significantly enhance current density and notably alter the I–V characteristics of GaN Schottky junctions. In the absence of an external load, GaN nanowires exhibit distinct rectifying behavior due to the presence of a Schottky barrier between GaN nanowires and Ag electrodes. Under bias voltage, when both ultraviolet light and mechanical load are simultaneously applied, there is a substantial increase in the current along with a decrease in turn-on voltage for the Schottky junction. This phenomenon can be attributed to combined actions involving piezoelectricity, photoinduced pyroelectricity, photoconductivity, and carrier screening effects. The photoconductivity effect leads to a generation of numerous photogenerated carriers through light excitation, which subsequently increases carrier concentration and enhances conductive ability within Schottky junctions. Furthermore, these photogenerated carriers possess a screening effect that effectively reduces the Schottky barrier. It should be noted that under constant pressure conditions, increasing ultraviolet light intensity gradually diminishes the effective Schottky barrier while augmenting the current (refer to Figure 5b). This provides a simple and effective method for controlling the current transmission of GaN devices by exploiting the coupling effect from pyro-phototronics to piezo-phototronics.

## 5. Conclusions

By coupling the pyro-phototronic and piezo-phototronic effects, we have investigated the ultraviolet and mechanical load-dependent electromechanical behaviors in a GaN nanowire. The main conclusions are drawn as follows:By combining with photoconductive, piezoelectric, pyroelectric, and photothermal effects, an optical–electromechanical coupling model of PSCs is proposed, and the universal analytical solutions of critical physical fields are obtained.Due to the synergistic effect of pyro-phototronic and piezo-phototronic phenomena, ultraviolet radiation significantly influences various physical field distributions in a GaN nanowire, including potential, polarization charge, and carrier concentration.The application of ultraviolet radiation can regulate the Schottky barrier height and electrical transport properties of GaN nano-Schottky devices. This provides a flexible and efficient method for modulating the electrical properties of GaN photovoltaic devices.

## Figures and Tables

**Figure 1 materials-16-06247-f001:**
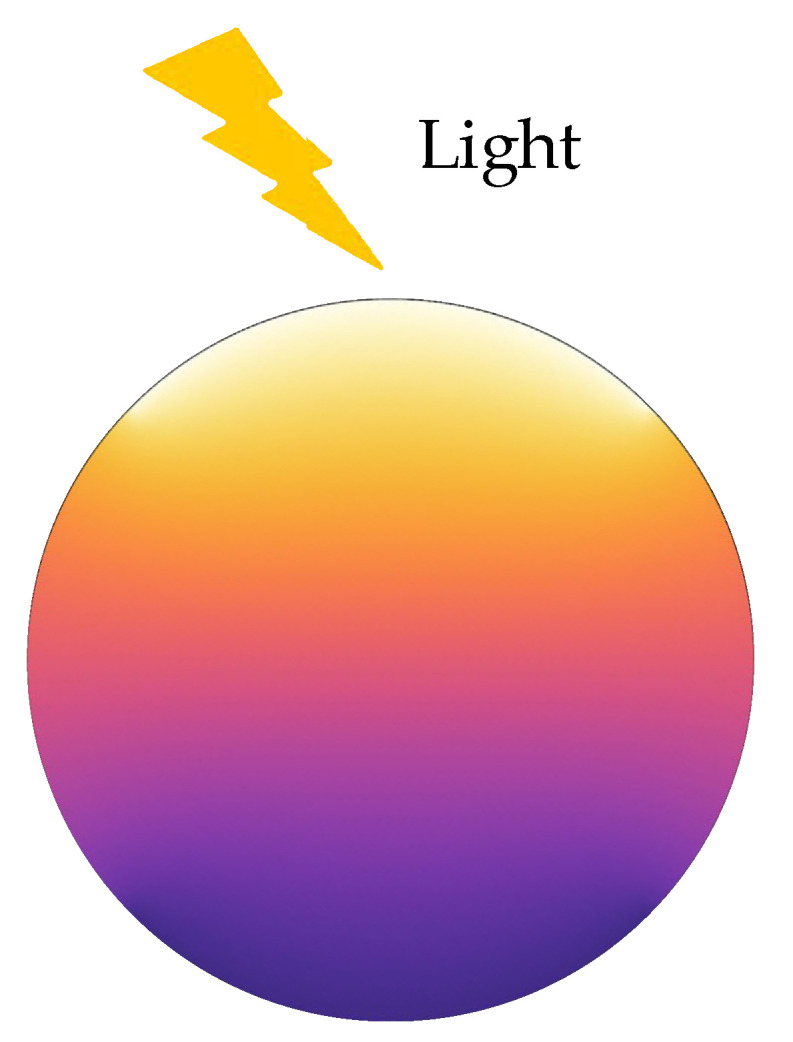
A schematic attenuation diagram depicting the decay of photo-excited carriers as the transmission depth increases under incident light.

**Figure 2 materials-16-06247-f002:**
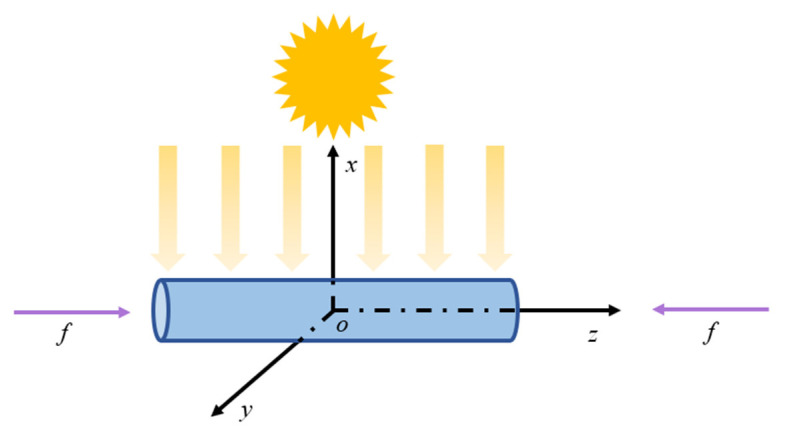
The schematic diagram of ultraviolet irradiation on a GaN nanowire (adapted from the ref. [26]).

**Figure 3 materials-16-06247-f003:**
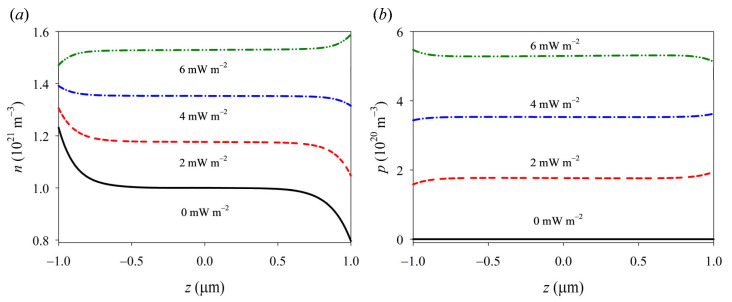
The carrier concentration distributions of (**a**) electrons and (**b**) holes with different ultraviolet intensities under compressive loading *f* = −2 MPa.

**Figure 4 materials-16-06247-f004:**
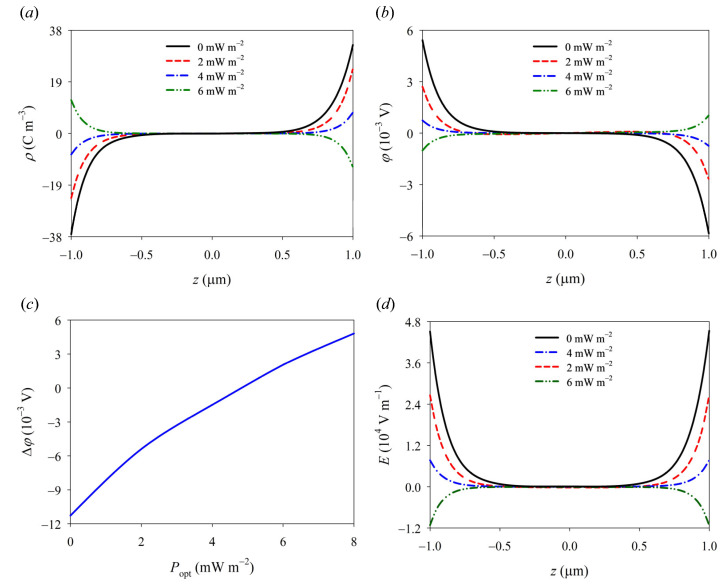
The physical field distributions of (**a**) the polarization charge density, (**b**) the electric potential, (**c**) the variation of the potential at the both ends, and (**d**) the electric field under a compressive stress *f* = −2 MPa.

**Figure 5 materials-16-06247-f005:**
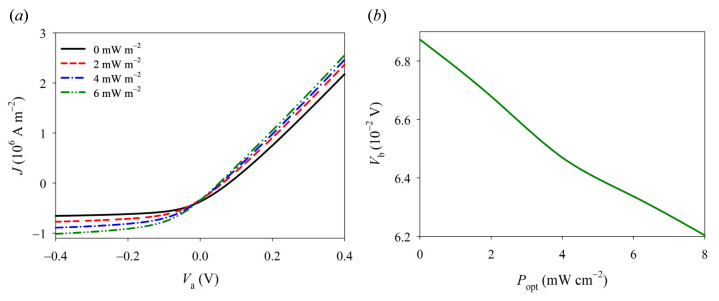
The features of (**a**) I–V and (**b**) *V*_b_–*P*_opt_ in a 1D GaN nanostructure under different ultraviolet intensities and a compressive stress *f* = −2 MPa.

## Data Availability

Data available on request due to restrictions, e.g., privacy or ethical. The data presented in this study are available on request from the corresponding author.

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
