# Peer review of "Coupling of Pyro–Piezo-Phototronic Effects in a GaN Nanowire"

_materials, 2023, doi:10.3390/ma16186247_

Round 1

Reviewer 1 Report

Authors should check the article before submitting it. Figures 3.4 and 5 do not have curves to verify the interpretation.

I think you need to resubmit the article to properly review it.

Author Response

We are very grateful to the reviewers’ comments that have led to further improvement of our manuscript. The main changes and modifications in the text have been highlighted with red font. We hope that the newly revised version has answered all or most of their criticisms.

Reviewer #1

Authors should check the article before submitting it. Figures 3.4 and 5 do not have curves to verify the interpretation.

Response: We apologize for our oversight. Problems related to these figures have been solved (see new Figures 3, 4 and 5).

Reviewer 2 Report

Dear Editor,
 thank you for sending me this manuscript to read, judge and comment on.
I apologise, but I downloaded the manuscript and opened it with different pdf software. Figures 3, 4, 5 appear to be blank panels with no curves or corresponding data points. Consequently, the article is unjustifiable for me.
Furthermore, it seems to me that the work relies heavily on the use of software based on the finite element method. Could the authors clarify whether their contribution is also focused on the development of the model? Or did they use mathematical models already available in the software packages?

Author Response

Reviewer #2

1) Figures 3, 4, 5 appear to be blank panels with no curves or corresponding data points. Consequently, the article is unjustifiable for me.

Response: We apologize once more for our oversight. We have carefully checked and replaced these Figures (see new Figures 3, 4 and 5).

2)It seems to me that the work relies heavily on the use of software based on the finite element method. Could the authors clarify whether their contribution is also focused on the development of the model? Or did they use mathematical models already available in the software packages?

Response: Following the suggestions, relevant clarification and explanation have been given in the revised version. Here it is worth noting that, this work mainly focuses on the analytical solution of a 1-D GaN nanowire under thermo-optoelectronic and piezoelectric photoelectron coupling. The numerical calculation was merely carried out for determining the I-V characteristics (see par. 2 on page 2).

Reviewer 3 Report

The article «Coupling of Pyro-Piezo-Phototronic Effects in a GaN Nanowire» by GuoShuai Qin et al is devoted to the study of the synergistic effect of ultraviolet and mechanical loading on the electromechanical behavior of a GaN nanowire. For study, the authors used a one-dimensional model that combines pyro-phototronic and piezo-phototronic properties. Based on the results obtained, the authors conclude that due to the pyro-phototronic effect and ultraviolet photoexcited non-equilibrium carriers, the electrical behavior of a nano-Schottky junction based on GaN can be modulated by ultraviolet light.

I suggest the authors pay attention to the following points of the article that need improvement:

1) In the "Introduction" section, it should be clearly justified why the GaN nanowire was chosen as the object for research.

2) In the "Introduction" section, the authors write that the actual scientific task in the field of research of piezoelectric semiconductors is to identify a coupling mechanism between mechanical, electric and optical fields in piezoelectric semiconductors. However, the article presents the results for only one type of piezoelectric semiconductors - GaN nanowires. Is it possible to generalize the conclusions made by the authors for other piezoelectric semiconductors? After all, it is well known that the response of a material to light exposure, as well as the transport properties of the material, depend significantly on the size of the band gap of the semiconductor. Is this fact taken into account in some way in the used model?

3) In the "Conclusion" section, the authors write that they have proposed an optical-electro-mechanical coupling model of piezoelectric semiconductors. Here it is necessary to briefly describe what are the elements of novelty of this model? How did the authors manage to connect all three types of fields within one model? Also, the authors should provide information about what is the criterion for the reliability of the proposed model? Has it been tested? If yes, how?

4) Figures 3, 4 and 5 do not display any curves in the pdf file of the article. Apparently, there was some kind of technical error while loading the article. There is no way to evaluate the calculation results obtained by the authors.

5) There is no mention of reference number 30 in the text of the article, although it is in the "Reference" section ("Theoretical Nanoarchitectonics of GaN Nanowires for Ultraviolet Irradiation-Dependent Electromechanical Properties"). This is another article by the authors of this paper, which already contains a lot of the information presented in the article presented in the review. In particular, it already describes one-dimensional (1-D) thermo-piezoelectric theory using GaN nanowires as an example. What then is the novelty of the author's new paper? There are quite a lot of repetitions of formulas between the two articles. It is necessary to clarify this point.

Minor editing of English language required.

Author Response

We are very grateful to the reviewers’ comments that have led to further improvement of our manuscript. The main changes and modifications in the text have been highlighted with red font. We hope that the newly revised version has answered all or most of their criticisms.

Reviewer #3

1)In the "Introduction" section, it should be clearly justified why the GaN nanowire was chosen as the object for research.

Response: As suggested, we have added a more detailed description in the revised section "Introduction" (see par. 2 on page 1).

2)In the "Introduction" section, the authors write that the actual scientific task in the field of research of piezoelectric semiconductors is to identify a coupling mechanism between mechanical, electric and optical fields in piezoelectric semiconductors. However, the article presents the results for only one type of piezoelectric semiconductors - GaN nanowires. Is it possible to generalize the conclusions made by the authors for other piezoelectric semiconductors? After all, it is well known that the response of a material to light exposure, as well as the transport properties of the material, depend significantly on the size of the band gap of the semiconductor. Is this fact taken into account in some way in the used model?

Response: Just as mentioned, this work mainly focuses on the analytical solution of a 1-D piezoelectric semiconductor. However, the mathematical model for coupling thermo-optoelectronic and piezoelectric photoelectrons is universal, which could be further generalized. This has been discussed and clarified in the revised version (see par. 3 on page 6).

3)In the "Conclusion" section, the authors write that they have proposed an optical-electro-mechanical coupling model of piezoelectric semiconductors. Here it is necessary to briefly describe what are the elements of novelty of this model? How did the authors manage to connect all three types of fields within one model? Also, the authors should provide information about what is the criterion for the reliability of the proposed model? Has it been tested? If yes, how?

Response: As suggested, we have provided more relevant information and highlighted the novelty of our model. As a preliminary study, the reliability and effectiveness of the proposed model were verified by using the numerical methods. The further work is still needed to elucidate the effect of such a complex and coupling field. This has been clarified in the revised version (see par. 2 on page 1, par. 2 on page 6, par. 1 on page 7 and par. 4 on page 9).

4)Figures 3, 4 and 5 do not display any curves in the pdffile of the article. Apparently, there was some kind of technical error while loading the article. There is no way to evaluate the calculation results obtained by the authors.

Response: We apologize for our oversight. These problems have been corrected (see new Figures 3, 4 and 5).

5) There is no mention of reference number 30 in the text of the article, although it is in the "Reference" section ("Theoretical Nanoarchitectonics of GaN Nanowires for Ultraviolet Irradiation-Dependent Electromechanical Properties"). This is another article by the authors of this paper, which already contains a lot of the information presented in the article presented in the review. In particular, it already describes one-dimensional (1-D) thermo-piezoelectric theory using GaN nanowires as an example. What then is the novelty of the author's new paper? There are quite a lot of repetitions of formulas between the two articles. It is necessary to clarify this point.

Response: Due to our negligence, Ref. 30 in the original manuscript was incorrectly referred to as 33, which might lead to confusion. Although some basic formulas such as the equilibrium equation are the same, this work mainly focuses on the analytical solution of a one-dimensional GaN nanowire under thermo-optoelectronic and piezoelectric photoelectron coupling, which is different from Ref. 30 as mentioned. This has been clarified in the revised version (see pars. 1 and 2 on page 2, pars. 2, 3 and 4 on page 6, and par. 4 on page 9).

Round 2

Reviewer 1 Report

The article is now in journal standards.

The authors should check Figure 1.

Reviewer 2 Report

The revised version of the manuscript has been improved with respect to the original submission. I would recommend acceptance of the manuscript.

Reviewer 3 Report

The authors took into account suggestions to improve the quality of the manuscript and made the necessary adjustments. I believe that the article can be published in the journal Materials.